# Current Understanding of Bacillus Calmette-Guérin-Mediated Trained Immunity and Its Perspectives for Controlling Intracellular Infections

**DOI:** 10.3390/pathogens12121386

**Published:** 2023-11-24

**Authors:** Ana Carolina V. S. C. de Araujo, Fábio Mambelli, Rodrigo O. Sanches, Fábio V. Marinho, Sergio C. Oliveira

**Affiliations:** 1Departamento de Genética, Ecologia e Evolução, Universidade Federal de Minas Gerais, Belo Horizonte 31270-901, MG, Brazil; carolvalente97@gmail.com; 2Departamento de Imunologia, Instituto de Ciências Biomédicas, Universidade de São Paulo, São Paulo 05508-900, SP, Brazil; fabio_mambelli@yahoo.com.br; 3Departamento de Bioquímica e Imunologia, Instituto de Ciências Biológicas, Universidade Federal de Minas Gerais, Belo Horizonte 31270-901, MG, Brazil; rcosanches@gmail.com (R.O.S.); fabiovitarelli@yahoo.com.br (F.V.M.)

**Keywords:** BCG, trained immunity, intracellular infections, innate memory

## Abstract

The bacillus Calmette–Guérin (BCG) is an attenuated bacterium derived from virulent *Mycobacterium bovis*. It is the only licensed vaccine used for preventing severe forms of tuberculosis in children. Besides its specific effects against tuberculosis, BCG administration is also associated with beneficial non-specific effects (NSEs) following heterologous stimuli in humans and mice. The NSEs from BCG could be related to both adaptive and innate immune responses. The latter is also known as trained immunity (TI), a recently described biological feature of innate cells that enables functional improvement based on metabolic and epigenetic reprogramming. Currently, the mechanisms related to BCG-mediated TI are the focus of intense research, but many gaps are still in need of elucidation. This review discusses the present understanding of TI induced by BCG, exploring signaling pathways that are crucial to a trained phenotype in hematopoietic stem cells and monocytes/macrophages lineage. It focuses on BCG-mediated TI mechanisms, including the metabolic-epigenetic axis and the inflammasome pathway in these cells against intracellular pathogens. Moreover, this study explores the TI in different immune cell types, its ability to protect against various intracellular infections, and the integration of trained innate memory with adaptive memory to shape next-generation vaccines.

## 1. Introduction

The bacillus Calmette–Guérin (BCG) vaccine, developed in the 1920s from virulent *Mycobacterium bovis*, has been administered for more than 100 years worldwide to protect children from severe forms of tuberculosis [1]. It is the only licensed vaccine for *Mycobacterium tuberculosis* (Mtb) control, given to around 100 million newborns annually [2,3]. Interestingly, the long-time usage of BCG in public health provided observations that drew attention to protective heterologous or non-specific effects (NSEs) [4,5]. For about 40 years, BCG has been the recommended intravesical immunotherapy for reducing the risk of tumor recurrence in patients with non-muscle invasive bladder cancer [6]. The BCG therapeutic potential has also been observed in the wart treatment arising from human papillomavirus (HPV) infection, in which patients revealed no recurrence among responders [7]. Moreover, epidemiological studies have shown that giving BCG to children is linked to an average decrease of 50% in infant mortality rates. This reduction is associated with an improvement in the initial immune response against sepsis, respiratory infections, and fever [8]. Other studies have conclusively shown that BCG revaccination significantly reduces the incidence of respiratory tract infections and pneumonia in both adolescents and the elderly [9,10]. Thus, evidence demonstrates that vaccinating humans with BCG can be beneficial against unrelated respiratory tract infections across a wide range of ages and can eliminate lesions, establishing a response similar to immunological memory and avoiding recurrence.

In essence, the BCG is a powerful tool in boosting the responsiveness of innate cells, particularly macrophages and dendritic cells (DCs), which function as antigen-presenting cells (APCs). The innate increased responsiveness leads to the activation of T cells and subsequent production of T helper cell type 1 (Th1) lymphocytes and interferon-gamma (IFN-γ) [11]. Overall, this inflammatory milieu contributes to an environment that favors cross-protection. Accordingly, the NSEs conferred by BCG against heterologous infections may originate from both innate and adaptive immunity responses. The heterologous adaptive immune response could occur by cross-reactivity (recognition of multiple antigens by T and B-memory cells) or bystander activation (activation of memory cells with weak or no costimulation by ongoing immune responses) [12]. Meanwhile, recent evidence emphasizes the importance of innate immune cells responding strongly during a subsequent, unrelated infection. Experiments using recombination activating gene *1* (RAG1) deficient or severe combined immunodeficiency (SCID) mouse strains (both lacking adaptive immune system) were of the utmost importance in this context [13,14,15,16], and encouraged further inquiries on how previous stimulation of innate immune cells could provide long-term cross-protection.

Deep research on the matter supported the conceptualization of a new innate immune system characteristic: trained immunity (TI). This term was coined by Netea et al. in 2011 and was well received by the scientific community in order to differentiate TI from the classical adaptive immune memory, historically referred to just as “immune memory” [17]. Among the key differences between TI and classical adaptive immune memory are (1) the maintenance of the memory status by epigenetic reprogramming instead of genetic rearrangement and selection of specific antigen receptors and (2) the lasting response provided by progenitor cells, mostly undifferentiated hematopoietic stem cells (HSCs), contrasting to immune experienced lymphocytes [13,14,15]. In summary, the TI is initiated by a primary stimulus that leads to epigenetic and metabolic reprogramming (induction phase). The organism or cells then return to their basal level after the initial stimulation is removed (resting phase). During a homologous or heterologous secondary challenge, an increased innate functional response is observed (trained response phase). [18,19,20]. The cross-protection provided by TI is one of its hallmarks.

The TI investigation exploiting BCG aroused defining important aspects in the field. For instance, the finding that HSCs can transmit the trained phenotype to derived innate cells was achieved using this vaccine as a TI-inducer [15]. This observation supports the presence of a sustained memory due to the transmission of epigenetic signatures from proliferative and long-lasting progenitors to effector cells [15,21]. Afterward, this scenario of HSC-protective imprinting was classified as central-trained immunity, differing from peripheral-trained immunity, which is observed in blood or tissue-resident cells (Figure 1) [22]. In the following sections, this review discusses evidence showing the central and peripheral TI mechanisms across different immune cell types, NSEs-mediated by BCG (including TI and T-lymphocyte heterologous responses) against intracellular pathogens, and detailed perspectives of how these mechanisms could be applied in the vaccine technology.

## 2. The Mechanisms of BCG-Mediated Trained Immunity in Multipotent Progenitors and Monocytic Lineage

As mentioned above, the BCG can reprogram innate cells, leading to their improved activity against related or unrelated pathogens. Accordingly, it has been demonstrated that BCG intradermally inoculated results in functional alterations of peripheral blood immune cells [23,24]. It was shown later that lasting (months or even years) peripheral trained responses due to intradermal BCG vaccination must be sustained by transcriptomic alterations in HSCs [25]. Additionally, BCG-systemic administration induces the reprogramming of HSCs in the bone marrow (BM) [15]. In these scenarios, TI-related mechanisms are dependent on shifts in metabolic circuits and epigenetic reprogramming [23]. Hence, metabolic and epigenomic rewirings in TI underlie functional changes, preparing cells and organisms to cope more readily when later exposed to different pathogens.

The central BCG-mediated TI is associated with the expansion of HSCs and their commitment to the myeloid lineage in the BM microenvironment [15]. Epigenetic changes induced by BCG in HSCs are indirect, as this bacillus is not able to infect these cells [15,26]. The BCG training of multipotent progenitors in BM is marked by increased STAT1 transcriptional levels, linked to a key role for IFN-γ since IFN-γ receptor (IFNGR)-deficient BM does not present expansion of HSCs upon intravenous (i.v.) BCG stimulation (Figure 1) [15]. Although IFN-γ seems to be critical in inducing BCG-mediated TI, it is not limited to it, as the full landscape of immune responses in BM needs to be assessed. This reasoning is well explained by a recent study by Khan et al., 2020, in which they show BM stimulation with Mtb leads to the activation of immune pathways resembling those induced by BCG, such as STAT1 and IFN-γ signaling. However, Mtb also strongly induces the type I IFN pathway, dysregulates iron metabolism, and depolarizes mitochondrial-membrane potential in a region of difference (RD-1) dependent manner. This scenario compromises myelopoiesis, resulting in the death of these cells by receptor-interacting protein kinase 3 (RIPK3)-induced necroptosis and triggers susceptibility to further challenges with Mtb. Meanwhile, immunization with β-glucan (another well-known TI-inducer) results in myelopoiesis induction in BM dependent on IL-1 pathway activation [27]. Thus, commitment to the myeloid lineage is a trait of the HSC-training process in BM, possibly induced by attenuated bacteria and pathogen-associated molecular patterns (PAMPs). Concordantly, the transfer of BCG-trained BM to immune-depleted mice confers greater antimicrobial control against heterologous infections compared to mice receiving naïve BM. It is feasible as trained effector cells will be derived from HSCs. These cells will inherit the epigenetic marks and migrate to the periphery during a heterologous infection (Figure 1). Hence, differentiated cells may show increased inflammatory responses in infected tissues [15,16].

The lower disease rate due to BCG vaccination has been associated with the H3K27ac enrichment (a histone marker observed in active promoters and enhancers) in regulatory regions of inflammatory cytokines in peripheral blood mononuclear cells (PBMCs) [10]. Accordingly, another study demonstrated that human peripheral blood monocytes, 30 days post-BCG vaccination, exhibited H3K27ac enrichment in several regions associated with the inflammatory response, highlighting the genes related to the PI3K/Akt pathway [23]. Furthermore, greater Akt/mTORC1/S6K pathway activation was observed in BCG-trained monocytes in vitro, being the mTOR phosphorylation required for a metabolic change characterized by increased glycolysis and glutamine pathway. Akt/mTORC1-pathway inhibition abrogated the H3K4me3 enrichment (a marker of activated regulatory regions) and the H3K27me3 reduction (a marker of inactivated regulatory regions) in promoter regions of inflammatory cytokines, such as IL-6 and tumor necrosis factor α (TNF-α) [20]. Given the above, the Akt/mTORC1-axis enhancement is a hallmark of BCG-trained monocytes, driving metabolic, epigenetic, and functional remodeling in these cells, which are integrated into an interdependent circuit (Figure 2). Additionally, a straight association between metabolism and epigenome was elucidated by the contribution of the glutamine pathway to chromatin modifications [20]. The glutamine pathway is responsible for metabolite accumulation in the tricarboxylic-acid (TCA) cycle, such as fumarate and citrate. These metabolites, after being transported from mitochondria to the cytoplasm, interfere with the histone-modifying enzymes [20,28].

Both BCG-trained human-circulating monocytes and mouse BM-derived macrophages (BMDMs) have a potentialized capacity to control pathogens, namely viruses, bacteria, protozoa, and fungi. The BCG-trained responses elicited by these cells are associated with higher control of Mtb, *Brucella abortus*, *Candida albicans*, *Leishmania braziliensis*, influenza A virus (IAV), yellow fever virus (YFV), and others [15,16,23,29,30]. These BCG-trained phagocytes present an increased expression of costimulatory molecules and secrete high levels of inflammatory cytokines, such as IL-6, TNF-α, IL-12, and IL-1β in response to unrelated stimulation (Figure 2) [16,20]. The improved ability to produce IL-1β appears to be a predictive factor of protection in BCG-mediated TI. For instance, trained human monocytes producing higher IL-1β amounts correlated with lower YFV viremia. Furthermore, genetic polymorphisms in the IL-1β promoter, IL-1 or IL-18 receptor coding gene, and inflammasome components, such as apoptosis-associated speck-like protein containing a CARD (PYCARD/ASC), are shown to affect the magnitude of BCG-trained responses, leading to lower production of other inflammatory mediators [23]. This evidence is supported by recent findings from our group, which demonstrated that canonical and caspase-11-mediated-non-canonical inflammasome pathways were enriched in BCG-trained BMDMs against *B. abortus* infection. The IL-1β release occurred along with the expressive caspase-1 processing, higher pro-IL-1β, caspase-11, NLRP3 expression, and GSDMD cleavage (Figure 2) [16]. Therefore, the IL-1 pathway plays a pivotal role in BCG-mediated TI, being by itself an inflammatory pathway associated with protection against viral and bacterial infections [16,23]. Certainly, higher NLRP3 expression by BCG-trained macrophages suggests the importance of this receptor in the TI establishment. Additionally, NLRP3’s role may be related to the mTORC1-complex activity, as both contribute to macrophage polarization towards the type 1 (M1) profile [31,32,33].

## 3. Trained Immunity Mechanisms in Neutrophils and Dendritic Cells (DCs)

Although most of the evidence linked to TI is demonstrated in macrophages and natural killer (NK) cells (subject well detailed in—[34,35,36]), BCG vaccination also induces the elevated functional capacity of peripheral blood neutrophils. Following 90 days post-vaccination with this bacillus, human-circulating neutrophils showed H3K4me3 enrichment in promoter regions of inflammatory cytokines (IL-1β and IL-8), mTOR, and phosphofructokinase (PFK, an enzyme that belongs to the glycolytic pathway). Despite presenting H3K4me3-enriched regulatory regions, chromatin accessibility did not reflect increased gene transcription compared to baseline. This finding suggests that neutrophils remain with more accessible gene regions, even though immune effector production only occurs upon a secondary stimulation [24]. This characteristic is similar to that previously observed in BCG-trained monocytes [23]. An exception to the immune effector production pattern was observed by higher PI3K/Akt-pathway expression upon BCG vaccination, indicating its crucial role in neutrophil training as well. Following in vitro secondary Mtb or *C. albicans* challenge, BCG-trained human neutrophils expressed an increased amount of activation markers, such as CD11b, and a lower quantity of programmed death-ligand 1 (PD-L1, a checkpoint protein associated with immunosuppression). Moreover, these trained cells presented superior degranulation capacity and increased production of reactive oxygen species (ROS), elastase, and lactate accompanied by elevated killing capacity against *C. albicans*, although neutrophil extracellular traps (NETs) production has not changed [24]. Therefore, BCG-trained neutrophils exhibited functional reprogramming based on epigenetic and metabolic changes, readily controlling the growth of heterologous pathogens through enhanced antimicrobial functions.

It is worth mentioning that BCG training in DCs has not been demonstrated so far. Nevertheless, the memory-development capacity of DCs has been reported following intranasal immunization of mice with an experimental strain of *C. neoformans* capable of producing IFN-γ (H99γ) [37]. DCs isolated from their lungs displayed increased transcriptional levels of IFN-γ, STAT1, IL-12, and nitric oxide synthase 2 (NOS2). Intriguingly, after 70 days of H99γ immunization, increased inflammatory-cytokine production by DCs from H99-γ-vaccinated mice was observed specifically against the *C. neoformans* challenge. Following *lipopolysaccharide* (LPS), *S. aureus*, and *C. albicans* exposure, these cells did not produce superior amounts of cytokines compared to controls. This evidence demonstrates a unique feature of training exhibited by DCs, suggesting a pathogen-specific response [37]. Another study, controversially, demonstrated that even upon exposure to immunomodulatory stimuli, such as IL-4, a type 1 DC profile induced by synergistic stimulation with IFN-γ and TNF-α in vitro was stable and resistant to switching to a type 2 DC profile [38]. H3K4me3 enrichment in promoter regions of inflammatory genes, like *Il12b* and *Nos2*, was also observed in the model. Moreover, the mixed lineage leukemia 1 (MLL1) methyltransferase enzyme mediated these epigenetic changes, and the presence of TNF-α was essential for stabilizing the inflammatory profile in DCs and their myeloid precursors [38]. These findings suggest broad applicability in the context of pathogens that possess immunomodulatory PAMPs responsible for impairing Th1 responses. Thus, whether DC-trained responses are pathogen-specific or broad-spectrum remains a point yet to be investigated. Moreover, whether BCG can induce TI in DCs against heterologous infections is still an open question.

## 4. Crosstalk between Trained Innate Immune Cells and Lymphocytes

It is supposed that the TI exhibited by innate cells interferes with adaptive immunity activity and vice versa. A piece of evidence is that the adoptive transfer of IFN-γ+TNF-α-trained DCs to TNF-α deficient mice previously infected with *C. neoformans* significantly raised the quantity of IFN-γ-producing T CD4^+^ lymphocytes while reducing the number of IL-5-producing CD4^+^ T lymphocytes [38]. Moreover, it is plausible that the increased production of inflammatory cytokines by BCG-trained macrophages, such as IL-12 and TNF-α, contributes to an environment that favors Th1 response establishment. Another example, demonstrated by our team and others, is that BCG-trained macrophages presented increased MHC-II receptor and costimulatory molecule expressions on the cell surface upon secondary challenge [16,39]. On a two-way street, BCG-trained macrophages can better control bacterial growth upon IFN-γ-exogenous stimulation compared to untrained macrophages. Accordingly, antimicrobial mechanisms exerted by macrophages, such as nitric oxide (NO) production, were enhanced in BCG-trained macrophages [16]. Even though the IFN-γ source can be from innate and adaptive immune systems, it is reasonable that IFN-γ produced by lymphocytes contributes to an increment in TI-effector functions. Further, recent studies have shown that neutrophils play an important role in the neighboring cells, including lymphocytes and macrophages [40,41]. Neutrophils may take part in inducing innate memory in macrophages. Chen et al., 2014, demonstrated that neutrophil depletion abrogated secondary protective responses exhibited by lung macrophages, which presented an elevated capacity to bind and kill pathogens [42]. Lastly, IFN-γ producing CD8^+^ T lymphocytes are indispensable for training alveolar macrophage in the airways [39]. It is still necessary to clarify which cellular interactions are essential in establishing TI and trained responses during heterologous infections. This premise must be considered for central and peripheral TI.

## 5. The BCG NSEs against Intracellular Pathogens Comprise TI and Heterologous T cell Responses

The BCG administration in infants results in a lower mortality rate during the first month after immunization, which is associated with protection against neonatal sepsis and respiratory infections [8,43]. Furthermore, BCG revaccination in the elderly leads to a reduction in the incidence of unrelated respiratory infections, mainly viral ones [10]. These NSEs have been associated with two main immunological mechanisms: TI and heterologous T cell immunity [44]. Kleinnijenhuis et al., 2013, by immunizing volunteers with BCG, identified the activation of both mechanisms. In this regard, cytokine production, resulting from stimulation of PBMCs with mycobacteria and unrelated pathogens, was evaluated up to one year after BCG immunization. Long-term effects were associated with heterologous Th1 and Th17 responses based on IFN-γ, IL-17, and IL-22 production one year post-immunization. Otherwise, the short-term effects were associated with TI up to three months after immunization, considering TNF-α and IL-1β production. Additionally, receptors of the innate immune response also had increased expression in monocytes, up to one year after BCG initial stimulus [45]. In another interesting study, Arts et al., 2018, used vaccination with the YFV as a heterologous infection model (as exemplified previously in Section 2). This is a suitable approach for investigating NSEs since viruses are detected in the circulation after vaccination, and changes in viremia would be essential to attest to the heterologous protection induced by BCG. Indeed, immunization with BCG 28 days before the YFV challenge resulted in a significant reduction in viremia. Moreover, functional alterations were evidenced by a significant increase in IL-1β and IL-6 production by PBMCs stimulated with LPS, Mtb, and *C. albicans*. Altogether, these findings experimentally certify the NSEs from BCG in humans [23].

Other studies demonstrated that BCG immunization promotes, in different mouse strains, significant protection against viral infections. They reported NSEs from BCG upon infection by IAV [46], Ectromelia virus [47,48], encephalomyocarditis virus [49,50], vaccinia virus (VV) [51], and others [52]. The protection conferred by BCG against encephalomyocarditis virus infection was correlated with macrophages in a lymphocyte-independent manner [49,50]. On the other hand, Mathurin et al., 2009, demonstrated that C57BL/6 mice showed resistance to VV when challenged 28 weeks after BCG immunization. The protective mechanism, however, does not appear to be related to TI but to the action of CD4^+^ T cells and IFN-γ. Notably, animals were subjected to antibiotic clearance before the challenge in this model [51].

Since the COVID-19 pandemic emerged, the use of BCG has been extensively discussed concerning SARS-CoV-2 infection [53,54]. At first, ecological studies raised the hypothesis that countries containing current national BCG vaccination policies for all demonstrate a lower incidence of mortality from COVID-19 [52]. Since then, several studies have explored the protective role of BCG immunization against COVID-19. Recently published studies bring contrasting findings on this issue. Kaufmann et al., 2022, reported that K18-hACE2 mice or Syrian golden hamsters that received BCG i.v. were as susceptible as the unvaccinated group to intranasal (i.n) or intratracheal (i.t) infection with SARS-CoV-2/SB2 (B.4 lineage) [29]. In turn, other studies demonstrated a different outcome [55,56]. Zhang et al., 2022, showed that i.v. BCG immunization of K18-hACE2 mice promotes a decrease in the pulmonary and upper airway viral load following i.n infection with WT SARS-CoV-2 [55]. High HSC proliferation in the BM, along with greater differentiation into myeloid cells and consequent enrichment of macrophages and DCs, supported TI. Furthermore, metabolic evaluation of plasma and PBMCs revealed greater activation of the glycolytic pathway in vaccinated animals [55]. Singh et al., 2022, demonstrated that i.v. BCG immunization in the Syrian golden hamster model elicited protection with reduced lung viral loads and bronchopneumonia. Single-cell transcriptome profiling demonstrated higher recruitment of plasma cells, Th1, Th17, regulatory T cells (Treg), cytotoxic T lymphocytes (CTLs), and memory T cells to the lungs of BCG-immunized mice. These findings were observed together with a transcriptional shift towards antigen presentation and tissue repair [56].

Additionally, Hilligan et al., 2020, reported BCG-mediated protection, when mice were i.v. immunized. Reduced viral loads, ameliorated lung pathology, and decreased cytokine production and inflammatory cell recruitment were observed in the model [57]. However, when mice were immunized subcutaneously (s.c.) with BCG, protection was not achieved. Finally, another study used a non-human primate model [58] where rhesus macaques were immunized with BCG by aerosol administration. Despite the quick induction of monocytes and γδ-T cells, overall protection was not achieved. As i.v. BCG immunization leads to direct engagement of the BM compartment (opposing local administration); one could suggest that TI induced in immune cell progenitors could account for stronger effects against viral challenge [54,59]. However, the mechanisms behind BCG i.v. immunization leading or not to protection against SARS-CoV-2 infection in the animal model still requires further elucidation.

As a parallel inquiry, several randomized trials are being conducted worldwide to evaluate whether BCG immunization is protective against COVID-19, but recent reports are still heterogeneous and not conclusive [54,59]. The trials, in general, evaluate the susceptibility and mortality against COVID-19 in countries in which neonatal BCG is a clinical practice and when adults are revaccinated with BCG. Here, several aspects could account for the reported heterogeneous observations. Firstly, immunological differences might be observed from BCG immunization of pre-vaccinated against BCG-naïve individuals. Secondly, the TI effects are expected to last up to 1–2 years [59], but additional influences should be studied throughout the lifespan. Also, susceptibility to infection should differ from healthcare workers to elders, from individuals with different genetic and environmental backgrounds, and also from individuals with comorbidities. All in all, despite these results raising several discussions, further investigation, such as broad meta-analyses, is required to establish a firm conclusion regarding BCG protection against COVID-19.

The NSEs mediated by BCG have also been reported against challenges by pathogens such as fungi and protozoa. SCID mice immunized with BCG and infected with *C. albicans* 2 weeks later presented a higher survival rate when compared to those immunized with PBS. This observation was accompanied by a reduction in the fungal burden in the kidneys. Furthermore, the same immunization-infection regimen generated splenic monocytes with greater capacity to produce TNF-α when stimulated with LPS [14]. The deficiency in T and B cells suggested the TI action as the protective mechanism in this context. This was reinforced by the observation that such protection is partially dependent on NK cells, which, in addition, produced higher IL-1β levels when stimulated by unrelated pathogens [60]. Concerning protozoa, Silva et al., 2021, demonstrated the relevant impacts of BCG in infections by different species of *Leishmania* spp. in vitro and in vivo. Training of human monocytes was performed with BCG for 24 h, followed by 5 days of resting and infection on the 6th day. The findings showed a significant reduction in the infection index of *L. braziliensis* and *L. infantum*. Additionally, mouse infection following 7 days of BCG i.v. administration promoted decreased *L. braziliensis* load in paw lesions and less *L. amazonensis* or *L. infantum* dissemination to organs such as the spleen and liver. The NSEs from BCG were mainly associated with the production of IL-32 [61]. BCG vaccination also affected the infection of another protozoan, *Plasmodium yoelii* (PyNL). Prior contact with BCG in this context promoted increased expression of antimicrobial molecules such as lactoferrin and cathelicidin, which were then associated with lower parasitemia observed in the animals [62].

Regarding the NSEs from BCG in bacterial infections, a recent study demonstrated that C57BL/6 animals immunized s.c. with BCG are protected against infection by *S. pneumoniae*. In addition to greater survival, animals had lower bacterial load in the lungs and spleen. Similar behavior was evidenced even when the infection was performed 16 weeks post-immunization. The observed protective phenotype was clearly related to the action of neutrophils and a possible contribution of alveolar macrophages [63]. Applying an experimental design slightly different, Mata et al., 2021, reported a heterologous protective effect against *S. pneumoniae*. However, in this approach, the protection and the reduced pathogen spreading were observed only due to BCG immunization by i.v. route [64]. Lastly, a study recently conducted by our team showed protective responses against *B. abortus* infection elicited 10 weeks after i.v. BCG immunization. We demonstrated that BMDMs presented a greater capacity to control intracellular bacterial replication along with potentiated production of inflammatory cytokines and metabolic rewiring. Importantly, we applied an antibiotic treatment following 4 weeks of BCG i.v. immunization, assuring that increased responses were mediated by TI and not due to concurrent stimulation. The reduction in the *B. abortus* burden in the spleen of C57BL/6 RAG^−/−^ mice also confirmed that the protection was associated with enhanced innate immune responses [16]. Given the above, it is clear that the NSEs from BCG have been consolidated over the decades, even though the TI-related mechanisms underlying such effects are still under elucidation. This clarification can be applied to the improvement of health policies, such as the development of vaccine strategies that take into account TI.

## 6. Trained-Immunity-Based Vaccines (TIbVs) and What It Adds to the Field of Vaccinology

Vaccines, as we know (i.e., conventional vaccines), are designed to evoke adaptive immune responses against a specific pathogen and promote long-lasting immune memory (Figure 3A). They are formulated to contain dead or attenuated pathogens, or even their subunits—purified or encoded by nucleic acids. B and T cell-specific responses will be orchestrated, and long-term protection will be marked by the persistence of circulating antibodies and immune memory cells, leading to reactivation upon new pathogen exposure [65]. However, pronounced immune responses might need immunizations containing adjuvants in their formulations for proper DC activation, especially for subunit and nucleic acid vaccines [66]. On the other hand, the so-called trained immunity-based vaccines (TIbVs) can be designed to work as activators of innate immunity, leading to a more responsive immune environment with higher resistance to a secondary insult (related or non-related) (Figure 3B). In essence, epigenetic reprogramming, metabolic alterations, and elevated cytokine production will lay the foundation for a broader-spectrum response, which will not only be beneficial against unrelated pathogens but also lead to an enhanced adaptive response upon subsequent immunizations with a specific antigen of interest. T and B cell proportions and functions (e.g., cell numbers, cytokine production, and proliferative rates) could be altered in this situation [67].

The TIBVs could prompt two scenarios. The first one (Figure 3B, enhanced non-specific response) was intensively discussed due to the COVID-19 pandemic when the use of BCG or Measles–Mumps–Rubella (MMR) vaccines had been postulated to be protective against SARS-CoV-2 infection [53,68]. The hypothesis goes by using a TI-inducer (such as the BCG vaccine) as a suitable ready-to-implement countermeasure in an epidemic-like situation, especially for the most vulnerable individuals, until vaccines that are more appropriate are developed. Here, TI-inducers could comprise the use of whole microorganisms (like bacteria, fungi, or viruses), purified molecules (like flagellins, lipopolysaccharides, or glucans), or even fully inactivated bacterial formulations (such as the MV30) [69,70,71]. However, licensed vaccines with proven heterologous effects against other pathogens would stand as more practical in order to achieve population, as safety and regulatory measures would have already been addressed. Nevertheless, not limited to epidemic situations, the study of licensed vaccines associations with protection against heterologous infections could significantly add to the health service’s arsenal against diseases. For instance, Lee et al., 2018, suggest that the cold-adapted live attenuated influenza vaccine (CAIV) could provide NSEs against unrelated respiratory pathogens [72]. Despite a growing body of reports on the field, unfortunately, most of them lack deep evaluation of the TI mechanisms underlying the observed phenotypes.

The second scenario (Figure 3B, trained immunity + enhanced specific responses) takes place in a more designed process. An immunization regimen could be conceptualized as a two-dose process (the priming dose consisting of a TI-inducer to potentialize the upcoming secondary immunization) or even mixing the inducer to a specific antigen. Leentjens et al., 2015, observed that immunization with the trivalent influenza vaccine (2013–2014 seasonal strains) 14 days after BCG immunization evoked significantly enhanced antibody responses marked by quicker seroconversion in immunized individuals [73]. Similar enhanced phenotypes were observed for vaccines designed against bacteria or their toxins (such as diphtheria, tetanus, and pneumococcal disease) when this BCG-training strategy was applied [74]. These findings highlight the importance of BCG immunization at birth, as it may work as a TI-based strategy for vaccines applied in infants [74,75]. As mentioned above, a methodological alternative for harnessing BCG adjuvant-like properties could be the administration of the antigen in association with the BCG in a one-dose immunization. Counoupas et al., 2021, demonstrated that a formulation containing BCG, the SARS-CoV-2 Spike protein, and Alum (there termed BCG:CoVac) was responsible for inducing elevated titers of neutralizing antibodies and Th1-biased cytokine response in K18-hACE2 mice [76]. This immunization regimen led to abrupt disease abrogation after the SARS-CoV-2 challenge, marked by a healthy clinical score of the mice, no body-weight loss, and no detectable viral titers in the lungs, with very few signs of tissue inflammation. Interestingly, when mice primed with BCG:CoVac was later boosted with a formulation containing only Spike and Alum, antibody responses were further augmented.

The previous approach has been investigated in another robust way: using a recombinant BCG (rBCG) strain able to express a specific antigen of interest. This methodology will combine the TI-inducer effects and the adjuvant-like properties of BCG with the adaptive-specific responses of a protein subunit vaccine in one single immunizer able to elicit strong cellular and humoral immune responses [77,78]. As the BCG replicates inside APCs, the initiation of the immune response can be improved, and its polarization may be directed by the BCG dose, as low doses are related to biased Th1 immune responses, while higher doses correlate with mixed Th1/Th2 immune responses [78]. Differently from the strategy of mixing BCG and antigen of interest, rBCG strains can be modified in order to evoke the best of the antigen immunogenicity. These adjustments can involve (1) mycobacterial signal sequences for driving antigen expression to the mycobacterium wall and enhancing recognition [79], (2) the use of stronger promoters for higher levels of antigen expression [80], or (3) the use of strains designed with the listeriolysin O from *Listeria monocytogenes*, a pore-forming protein which leads to leakage of the mycobacterium antigens, enhancing antigen processing and presentation by the host [81].

In a strategy for an experimental model of pertussis, Nascimento et al., 2000, evaluated a rBCG strain designed to express a genetically detoxified S1 subunit from a pertussis toxin (S1PT) in fusion with a β-lactamase signal sequence (rBCG-S1PT) under the control of a strong promoter [82]. They reported that mice immunized with rBCG-S1PT presented a Th1-dominant immune response, being highly protected against intracerebral challenge with *Bordetella pertussis*. In a different strategy for an experimental model of leprosy, Ohara et al., 2001, assessed a rBCG strain overproducing three components against mycobacterium infection (rBCG/BA51) [83]. They observed that immunization of C57BL/6 and BALB/c mice led to reduced multiplication of the intracellular bacteria *Mycobacterium leprae* in their foot pads and elevated IFN-γ levels in their spleens (when cells were stimulated with *M. leprae* lysate). Not limited to bacterial infections, the rBCG system could be applied to other sorts of antigens, ranging from viral [84] to parasitic [85] molecules with elevated immune results.

Our group has recently demonstrated how combining all the immune features the rBCG system disposes of could potentiate protection against SARS-CoV-2 [86] in the K18-hACE2 murine model. We demonstrated mice immunized with rBCG-ChD6 (an rBCG strain expressing a chimeric protein based on immunodominant epitopes from Nucleocapsid and Spike proteins from SARS-CoV-2) followed by a booster dose containing the purified chimeric protein (rChimera), and Alum presented the best protective phenotypes. This immunization regimen induced the highest levels of antibodies against rChimera, neutralizing antibodies against SARS-CoV-2 and cytokine responses (IFN-γ and IL-6). These results were accompanied by IgG2c/IgG1 isotype switch, no body-weight loss in mice, and no detectable viable virus in their lungs. However, mice immunized with wild-type BCG followed by a booster dose with rChimera associated with Alum presented lower antibody levels, deteriorated clinical phenotype, and significantly higher viral load in their lungs (assessed by qPCR). When mice did not receive any BCG immunization, the detected cellular responses were correspondingly lower. Additionally, a double-dose rBCG-ChD6 immunization (no protein-Alum boost) was not protective. Taken together, these results showed that rBCG expressing target infection-related antigens is a more robust strategy for priming than wild-type BCG in this scenario. Further, rBCG immunization associated with standard specific antigen representation in a priming-boost strategy was crucial to significantly enhanced protection against SARS-CoV-2 infection. In conclusion, TIbVs represent a conceptually new field for vaccine design with a variety of possibilities to be explored. The upgrowing body of investigative research on TI mechanisms also highlights the importance of the clinical evaluation of existing vaccines regarding NSEs. Elucidating its mechanisms could aid translational studies, resulting in the rational design of more robust and protective vaccines for the population.

## 7. Concluding Remarks and Future Directions

The BCG is the most commonly administered vaccine in human history. Traditionally, this attenuated bacterium is used to prevent severe forms of tuberculosis in children. Furthermore, multiple studies have confirmed that this vaccine is indeed capable of triggering non-specific preventive or therapeutic beneficial effects against unrelated pathogens and cancer [6,16,87]. These non-specific BCG-mediated effects have been applied in clinical practice for about 40 years, consisting of the gold-standard intravesical therapy recommended to reduce the recurrence risk of non-muscle invasive bladder cancer after transurethral resection [6]. The mechanisms of bladder cancer treatment using BCG are currently under research. Recent findings suggest an important role for the adapter molecule MyD88 in association with a redundant function of Toll-like receptors (TLRs) 3, 7, and 9 [87].

Both innate and adaptive immune systems play a crucial role in producing heterologous effects. Trained immunity (TI), or innate immune memory, which is mainly demonstrated in myeloid and NK cells, has been shown to elicit an important role in BCG cross-protection against unrelated infections [6]. Moreover, TI can help manage nosocomial infections, such as skin infections and pneumonia. In mouse models, it was shown improved response to methicillin-resistant *Staphylococcus aureus*, *Acinetobacter baumanii*, multidrug-resistant *Pseudomonas aeruginosa*, and *Klebsiella pneumoniae* [88,89]. Therefore, exploring BCG can expand our understanding and application of TI in the clinical setting, which can ultimately lead to an improved quality of life for people.

The BCG-induced epigenetic and metabolic alterations were previously demonstrated in monocytes/macrophages, NK cells, and neutrophils. In addition, BCG also changed γδ T cells’ transcriptional programs and increased their responsiveness to heterologous bacterial and fungal stimuli, including LPS and *Candida albicans* [90]. More immune cell types are being involved in the process of TI, and its applications are increasing day by day. Finally, TI comprehension, including signaling pathways, cell-to-cell interactions, and durability, can be used to develop the next generation of vaccines to prevent or treat various infections and orphan diseases. This review discusses the mechanisms and protective role of TI against intracellular pathogens, focusing on integrating trained innate memory with adaptive memory to shape next-generation vaccines. Two important points for future direction are: (i) to determine new immune cell subpopulations involved in TI, spotlighting their activation process, and (ii) to identify new pathogens or their pathogen-associated molecular pattern as inducers of innate memory.

## Figures and Tables

**Figure 1 pathogens-12-01386-f001:**
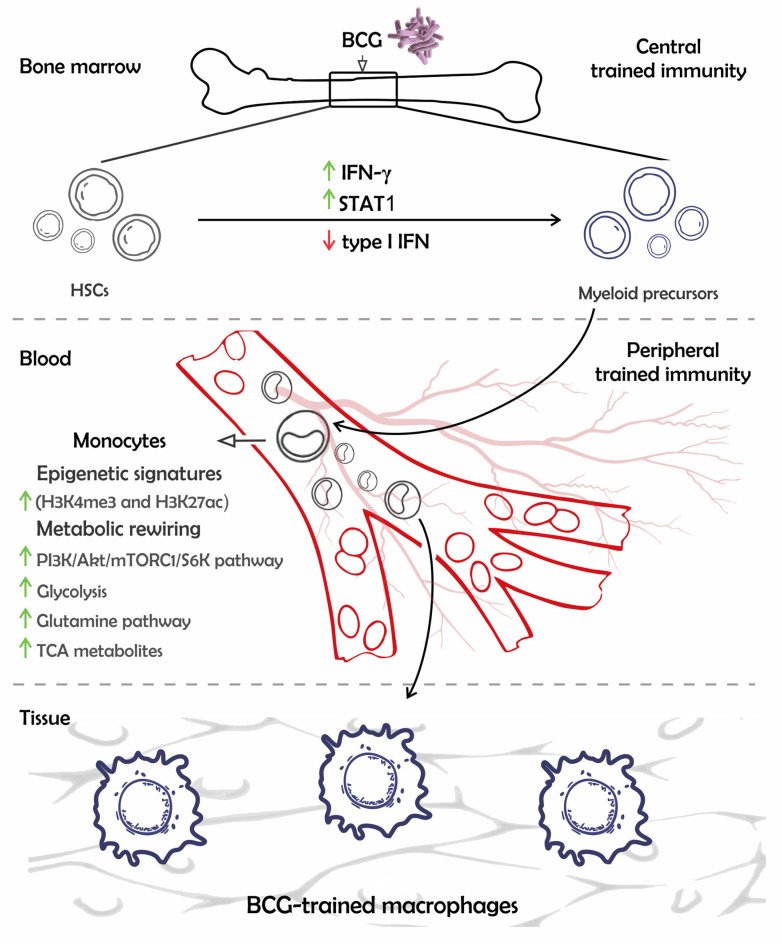
BCG-trained HSCs differentiate into monocytes and macrophages presenting epigenetic and metabolic rewirings. The BCG stimulates HSCs in the bone marrow (BM) microenvironment, inducing STAT1 and IFN-γ signaling. These pathways associated with epigenetic alterations in HSCs lay the foundation for BCG-induced central TI. The training of these cells is the explanation for long-term responses due to effector cells differentiated from HSCs, such as monocytes and macrophages, inheriting chromatin modifications and being prone to responding effectively upon unrelated infections. Differentiated cells also harbor metabolic changes, which help to maintain the epigenetic marks and immunologic enhancement. The understanding of mechanisms related to trained monocytes in blood represents the peripheral TI. [Signal transducer and activator of transcription 1 (STAT1), Interferon-gamma (IFN-γ), Histone 3 lysine 4 methylation (H3K4me3), Histone 3 lysine 27 acetylation (H3K27ac), *Mammalian target of rapamycin complex 1* (mTORC1)*,* and Tricarboxylic-acid (*TCA*)].

**Figure 2 pathogens-12-01386-f002:**
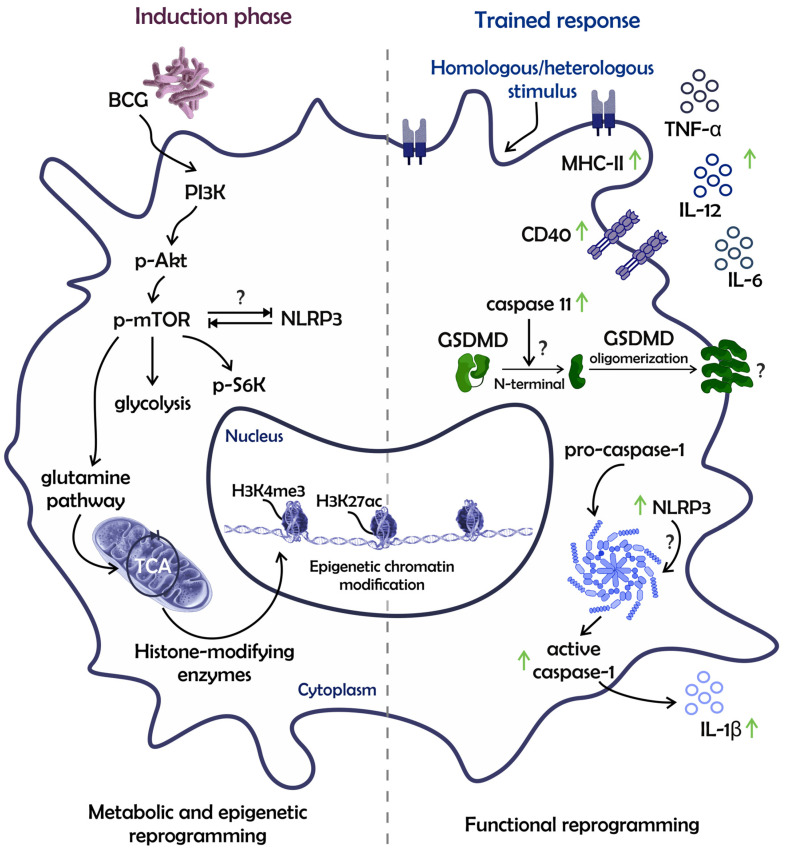
Metabolic, epigenetic, and functional remodeling in BCG-trained macrophages are integrated into an interdependent circuit. The BCG induces metabolic and epigenetic alterations (left) that enable these phagocytes to perform functionally improved responses following stimulation (right). The metabolic shift includes enhancement of the PI3K/Akt/mTORC1 axis, which in turn orchestrates other changes in the metabolism, such as S6K phosphorylation, induction of glycolysis, and glutamine pathway. The last one has a key role in TCA-metabolite accumulation and epigenetic chromatin modifications. Furthermore, there is a higher NLRP3 expression in BCG-trained macrophages when compared to untrained counterparts. The possible crosstalk between mTOR and NLRP3 still needs to be demonstrated in BCG-mediated TI. All these cellular changes contribute to an improved response following a heterologous stimulus. Upon stimulation, these trained cells could exhibit increased expression of costimulatory molecules on the cell surface, inflammatory cytokine release, and inflammasome activation. The non-canonical and canonical inflammasome platforms could be enhanced in BCG-trained macrophages, contributing to IL-1β secretion. (The interactions marked by the symbol—?—require formal elucidation). [Phosphoinositide-3-kinase (PI3K), *Protein kinase B* (Akt), Mammalian target of rapamycin (mTOR), Ribosomal protein S6 kinase (S6K), *NOD-like receptor (NLR) family pyrin domain-containing 3* (*NLRP3*)*,* Gasdermin D (GSDMD), *Major histocompatibility complex* class II (MHC-II), Cluster of differentiation (CD), Tumor necrosis factor α (TNF-α), and Interleukin (IL)].

**Figure 3 pathogens-12-01386-f003:**
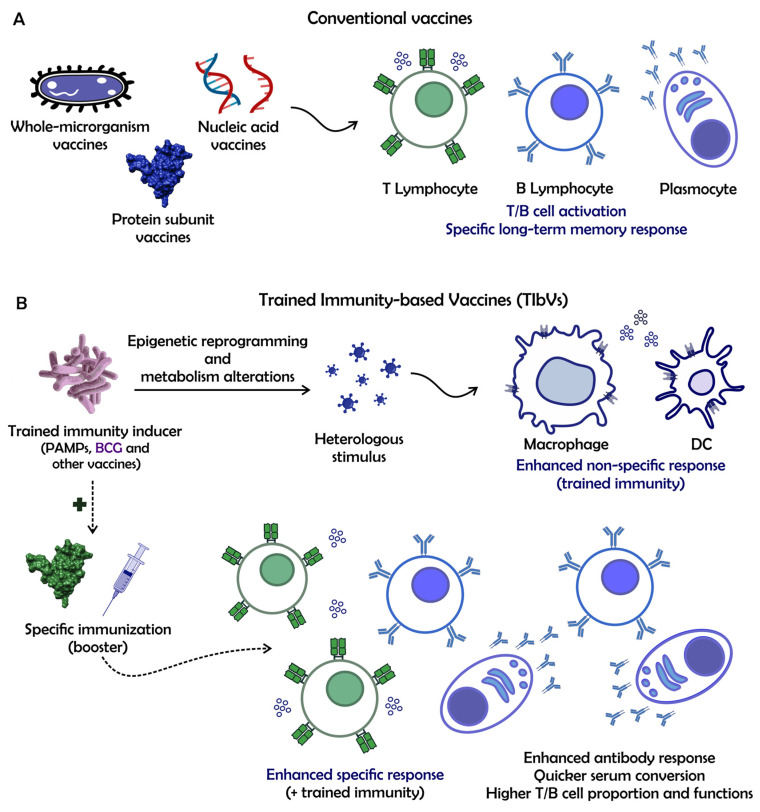
Conventional and trained immunity-based vaccine approaches. Schematic representation of (**A**) conventional vaccine and (**B**) trained immunity-based vaccine (TIbV) designs. (**A**) Conventional vaccines are designed to achieve T/B cell activation and specific long-term memory responses against a specific pathogen. (**B**) On the other hand, by making use of a TI inducer (PAMPs such as glucans or vaccines known to induce TI), TIbVs aim to stimulate the innate immune system (straight arrows). Epigenetic reprogramming and metabolism alterations will lead to (upon a secondary stimulus) an enhanced non-specific response orchestrated by the innate immune system. Moreover, this method can also be used to improve the adaptive immune response (dashed arrows) when the TI inducer is followed by a booster immunization with a specific antigen. The more responsive innate immune system will lead to an enhanced adaptive response, with elevated humoral responses, quicker serum conversion, and higher T/B cell proportions and functions.

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
