# Peer review of "Current Understanding of Bacillus Calmette-Guérin-Mediated Trained Immunity and Its Perspectives for Controlling Intracellular Infections"

_pathogens, 2023, doi:10.3390/pathogens12121386_

Round 1

Reviewer 1 Report

Comments and Suggestions for Authors

The manuscript entitled, “Current understanding of bacillus Calmette-Guérin-mediated trained immunity and its perspectives for controlling intracellular infections” is a well-organized and detailed presentation of “trained immunity”. This review is a new area of investigation in the field of immunology and, therefore, presents novelty to the field. Interesting aspects of the review are the coverage of numerous cell types including macrophages, dendritic cells, neutrophils, and natural killer cells that can express trained immunity. These cell types are traditionally considered components of innate immunity that do not alter their phenotype during infection. The review provides comprehensive examples of trained immunity and mechanisms to explain how training occurs. Documentation of trained immunity in different gene knock-out mice where adaptive immunity is deficient supports the scientific soundness of this novel field. The figures are appropriate to provide a visual description of differentiation from hematopoietic stem cells to trained monocytes and macrophages. BCG has long been known to have beneficial indirect immunological effects in multiple treatments from enhancing immune responses to various vaccines to cancer therapy; however, the mechanisms have been historically unclear. The present review provides scientific soundness to aid in understanding how trained immunity may play a central role in the BCG indirect effect to enhance cross-talk between innate and adaptive immunity. The mechanistic understanding of how trained immunity functions based on the present review provides potentially new approaches and justifications in traditional and nontraditional vaccines.

There are two minor typographical errors.

1.     Figure 3, panel B, “Trained mimmunity”….

2.     Line 487- “immun”…

Comments on the Quality of English Language

There are no issues with the quality of English. The manuscript is well written. Small typos are noted.

Author Response

Reviewer#1

The manuscript entitled, “Current understanding of bacillus Calmette-Guérin-mediated trained immunity and its perspectives for controlling intracellular infections” is a well-organized and detailed presentation of “trained immunity”. This review is a new area of investigation in the field of immunology and, therefore, presents novelty to the field. Interesting aspects of the review are the coverage of numerous cell types including macrophages, dendritic cells, neutrophils, and natural killer cells that can express trained immunity. These cell types are traditionally considered components of innate immunity that do not alter their phenotype during infection. The review provides comprehensive examples of trained immunity and mechanisms to explain how training occurs. Documentation of trained immunity in different gene knock-out mice where adaptive immunity is deficient supports the scientific soundness of this novel field. The figures are appropriate to provide a visual description of differentiation from hematopoietic stem cells to trained monocytes and macrophages. BCG has long been known to have beneficial indirect immunological effects in multiple treatments from enhancing immune responses to various vaccines to cancer therapy; however, the mechanisms have been historically unclear. The present review provides scientific soundness to aid in understanding how trained immunity may play a central role in the BCG indirect effect to enhance cross-talk between innate and adaptive immunity. The mechanistic understanding of how trained immunity functions based on the present review provides potentially new approaches and justifications in traditional and nontraditional vaccines.

There are two minor typographical errors.

1- Figure 3, panel B, “Trained mimmunity”….

Answer: Thank you for the spelling correction. It was changed on panel B from Figure 3.

2- Line 487- “immun”…

Answer: It was corrected and it is yellow highlighted in the manuscript, section: concluding remarks and future directions, currently line 563.

Reviewer 2 Report

Comments and Suggestions for Authors

The authors nicely framed the manuscript on the long-lasting immunity of BCG vaccination on trained immunity and heterologous immunity. The manuscript deeply quoted the TI molecular mechanisms, signaling pathways and also elucidate the metabolic reprogramming and epigenetic modifications. However, there are a variety of points that should be addressed.

Major

1.      Firstly, the organization and content of the abstract section are not sufficiently clear and

concise. A brief and concise summary is needed to convey the overall picture of the review.

2.      In the introduction section, it is suggested to rationalize and organize the presentation of the background of BCG introduction. It should flow naturally from the background to the focal points of the review.

3.      Authors should add the role of BCG trained immunity in the context of Covid-19 under the section of BCG NSEs against intracellular pathogens. A thorough literature search on this topic is required to take published knowledge into account. I would recommend this paper (PMID: 32393823).

4.      The application of BCG induced TI to be explored and necessary to clarify the effect of trained immunization on various diseases including Cancer and neuro degenerative diseases. I didn’t find much discussion on BCG induced TI to develop anti therapeutics in infectious diseases and antitumor therapeutics. I will ask Authors to discuss briefly about the potential applications of BCG-induced trained immunity as therapeutic options. (PMID: 32393823)

5.      Finally, it is recommended to improve the resolution of the images.

Comments on the Quality of English Language

The grammar in the article is not sufficiently smooth, paying attention to the coherence of transitions.

Author Response

Reviewer#2

The authors nicely framed the manuscript on the long-lasting immunity of BCG vaccination on trained immunity and heterologous immunity. The manuscript deeply quoted the TI molecular mechanisms, signaling pathways and also elucidated the metabolic reprogramming and epigenetic modifications. However, there are a variety of points that should be addressed.

Major

1- Firstly, the organization and content of the abstract section are not sufficiently clear and concise. A brief and concise summary is needed to convey the overall picture of the review.

Answer: We appreciate your suggestion and agree with you that the abstract can benefit from being more detailed and well-organized. Therefore, we have restructured it accordingly. In the current version, we believe the abstract is both concise and provides a clear overview of the review. We ensured that all the required changes were made while adhering to the Pathogens guidelines, which require a maximum of 200 words. It is yellow highlighted in the manuscript.

2- In the introduction section, it is suggested to rationalize and organize the presentation of the background of the BCG introduction. It should flow naturally from the background to the focal points of the review.

Answer: Thank you for your suggestion. The introduction now starts with a background on BCG and leads into the main points discussed in the review. To respond to this requirement, we have added two paragraphs to the introduction, which are yellow-highlighted.

3- Authors should add the role of BCG-trained immunity in the context of Covid-19 under the section of BCG NSEs against intracellular pathogens. A thorough literature search on this topic is required to take published knowledge into account. I would recommend this paper (PMID: 32393823).

Answer: Thank you for your input. We reformulated the topic and added a discussion regarding BCG-induced TI in the context of COVID-19 under the section on BCG NSEs against intracellular pathogens. We expanded the discussion to address different findings over animal models and randomized trials. It is yellow highlighted for your convenience, lines 318-323 and 332-364.

4- The application of BCG-induced TI to be explored and necessary to clarify the effect of trained immunization on various diseases including Cancer and neurodegenerative diseases. I didn’t find much discussion on BCG-induced TI to develop anti-therapeutics in infectious diseases and antitumor therapeutics. I will ask Authors to discuss briefly about the potential applications of BCG-induced trained immunity as therapeutic options. (PMID: 32393823)

Answer: Despite the clinical use of BCG in non-muscle invasive bladder cancer therapy and published studies involving its therapeutic application in melanoma, neurodegenerative diseases, and warts, it is not the main subject discussed in this review. Thus, we discussed briefly the therapeutics involving BCG in cancer, warts, and infectious diseases in the sections: introduction and concluding remarks and future directions. It is yellow-highlighted in the current manuscript, lines 40-46 and 533-552.

5- Finally, it is recommended to improve the resolution of the images.

Answer: The final images are at 300 dpi resolution. However, when they were attached to the final manuscript, the resolution decreased greatly. This issue has now been resolved through changes to document settings.

Reviewer 3 Report

Comments and Suggestions for Authors

The authors provide a much-needed review describing the trained immunity mediated by the bacillus Calmette-Guérin (BCG).

The authors make a comprehensive view of the trained-immunity mechanisms across different cell types, describing the non-specific effects mediated by BCG, including also the heterologous responses, and how these mechanisms may be applied in the vaccine technology.

The manuscript is well written and appropriately summarizes the recent works regarding the topic of the review. The figures are clear and are helpful to better understand the text.

 Please find below just few comments:

-          Some abbreviations (e.g. GSDMD, i.v, i.c….) have not been reported extensively. Please double-check the text.

-          In the legend of figure 2 replace IL-β with IL-1β.

-          Line 487: Replace “impotart” with “important”.

Author Response

Reviewer#3

The authors provide a much-needed review describing the trained immunity mediated by the bacillus Calmette-Guérin (BCG). The authors make a comprehensive view of the trained immunity mechanisms across different cell types, describing the non-specific effects mediated by BCG, including also the heterologous responses, and how these mechanisms may be applied in the vaccine technology. The manuscript is well-written and appropriately summarizes the recent works regarding the topic of the review. The figures are clear and are helpful to better understand the text.

Please find below just few comments:

  1. Some abbreviations (e.g. GSDMD, i.v, i.c….) have not been reported extensively. Please double-check the text.

Answer: Thank you for your suggestion. We have carefully checked the acronyms used throughout the text and made sure to include the full names of molecules when necessary. The modifications we made have been highlighted in yellow for your convenience. In some cases, we have added the names to the legend of Figures 1 and 2 where multiple acronyms are referenced. This was done because Topics 2 and 3 contain numerous acronyms, which would have resulted in truncated text if we had written them all in full. For all other modifications, you can find the details in the main body of the text.

  1. In the legend of Figure 2 replace IL-β with IL-1β.

Answer: It was replaced on the legend from Figure 2, line 208.

  1. Line 487: Replace “impotart” with “important”.

Answer: It was corrected and it is yellow highlighted in the manuscript, section: concluding remarks and future directions, line 562.

Round 2

Reviewer 2 Report

Comments and Suggestions for Authors

I am satisfied with the revised version. All of my comments were addressed nicely.